# Nano-Hydroxyapatite (nHAp) in the Remineralization of Early Dental Caries: A Scoping Review

**DOI:** 10.3390/ijerph19095629

**Published:** 2022-05-05

**Authors:** Aiswarya Anil, Wael I. Ibraheem, Abdullah A. Meshni, Reghunathan S. Preethanath, Sukumaran Anil

**Affiliations:** 1Saveetha Dental College, Saveetha Institute of Medical and Technical Sciences (SIMATS), Chennai 602105, India; aiswaryanil@gmail.com; 2College of Dentistry, Jazan University, P.O. Box 114, Jazan 45142, Saudi Arabia; wibraheem@jazanu.edu.sa (W.I.I.); ameshni@jazanu.edu.sa (A.A.M.); drpreethanath@gmail.com (R.S.P.); 3Oral Health Institute, Department of Dentistry, Hamad Medical Corporation, Doha 3050, Qatar; 4College of Dental Medicine, Qatar University, Doha 2713, Qatar

**Keywords:** nano-hydroxyapatite, dentifrices, scoping review, dental caries, remineralization, demineralization

## Abstract

(1) Background: Nano-hydroxyapatite (nHAp) has been reported to have a remineralizing effect on early carious lesions. The objective of this scoping review was to analyze the remineralization potential of nano-hydroxyapatite (nHAp)-containing dentifrices, by mapping the existing literature. (2) Methods: This review was performed using the PRISMA-ScR Checklist, which is an extension of the PRISMA Checklist for Systematic Reviews and Meta-Analyses. In this study, the population, concept, and context (PCC) framework was used to find relevant papers published between 2010 and 2021. Nano-hydroxyapatite (nHAp) and dentifrices containing nHAp as one of the ingredients were the two main concepts of the research question. MeSH phrases, keywords, and other free terms relevant to nano-hydroxyapatite and dentifrices were used to search the literature databases. (3) Results: Preliminary searches yielded 59 studies; the title and abstract screening results excluded 11 studies. The remaining studies were thoroughly reviewed by two reviewers on the basis of the inclusion and exclusion criteria. Finally, 28 studies were included, and 20 studies were excluded. Most of the studies that were included reported that when nHAp was used alone, it had many different effects, such as remineralization, caries prevention, less demineralization, brighter teeth, less pain, and remineralization of enamel after orthodontic debonding. (4) Conclusions: Dentifrices that contain nHAp offer a variety of therapeutic and preventative effects. Currently, there is insufficient evidence to support the efficacy of nHAp dentifrices in primary teeth. Additional long-term investigations using standardized protocols are required to reach decisive conclusions about the effects of nHAp dentifrices on primary and permanent dentitions.

## 1. Introduction

Dental caries is the most common oral illness, affecting people of all ages [1]. They causes discomfort, impair functionality, and degrade one’s quality of life [2]. Additionally, the high cost of treating dental caries is an economic burden on both individuals and healthcare systems [3]. Numerous variables have been implicated in the demineralization of enamel and dentine, including cariogenic bacteria, changes in salivary pH, and fermentable carbohydrates. Dental caries damage the surface of teeth and progress through stages of demineralization and remineralization before invading deeper levels of tooth lesions [4]. Similarly, during orthodontic bonding and debonding, an uneven enamel surface with micro-damage and white spot lesions is frequently observed [5], which also contributes to the likelihood of demineralization [6]. A gradual decrease in prismatic mineral content (critical layer) is the hallmark of the early stage of enamel demineralization [4]. This sluggish process can be corrected by detecting enamel caries early and applying remineralizing chemicals.

Fluoride at a concentration of 1000–1450 parts per million has been utilized as a preventive and remineralization agent due to its ability to promote the development of fluorapatite [7]. For years, brushing your teeth regularly with a fluoride dentifrice has been recommended; however, there is a possibility of fluoride poisoning and fluorosis in children, particularly those under the age of six, as a result of continuous use of fluoride at higher concentrations [8]. Additionally, an acid-resistant layer can form that prevents the diffusion of remineralizing ions into deeper layers, thereby limiting remineralization across the area [9]. Moreover, it has been observed that high concentration fluoride dentifrices can have a remineralizing effect on carious lesions and can reduce demineralization; however, to date, there is no compelling evidence that these are more useful than the standard concentration dentifrices [10]. Calcium phosphate-based treatment has also been identified as a possible alternative remineralizing agent with anti-caries effects. Casein phosphopeptide-amorphous calcium phosphate is one such agent (CPP-ACP). It has been shown to suppress demineralization by releasing calcium and phosphate ions in low pH conditions, such as those present in carious lesions; however, there are only a few studies that support its use [11].

Nano-hydroxyapatite (nHAp) has received considerable attention in recent years for its use in a variety of preventative, therapeutic, and regenerative therapies. Hydroxyapatite (HA) is a mineral that has been extensively employed in periodontal bone regeneration, tissue engineering, and dentinal hypersensitivity, as well as being used as a remineralization agent [12]. Tooth enamel is primarily composed of HA crystals ranging in size from 20 to 40 nm. When these particles mature, they solidify, limiting their ability to undergo biological remodeling during demineralization. Thus, synthetic nHAp has begun to be employed for remineralization purposes and has gained prominence due to the structural and chemical resemblance of nano-sized HA crystals to enamel apatite crystals [12,13]. Additionally, it is more biocompatible, and has stronger bioactivity, resorption, and mechanical qualities than HA; nHAp has been shown in studies to have a remineralizing effect on artificial carious lesions and to build a new enamel layer [2,14,15,16,17]. Due to nHAp’s remineralizing ability on enamel, some experts believe it is a bionic material capable of regenerating enamel [5,18].

## 2. Materials and Methods

### 2.1. Research Question

This review is reported based on the Preferred Reporting Items for Systematic reviews and Meta-Analyses extension for Scoping Reviews (PRISMA-ScR) Checklist [19]. The review protocol was prepared prior to commencement and was based on existing best practices [20,21]. The research question for this scoping review was “Does the addition of nHAp in dentifrices have a positive effect on the remineralization of enamel?” The purpose of this study was to investigate and map the evidence on the remineralization potential of nano-hydroxyapatite (nHAp)-containing dentifrices by mapping the existing literature.

### 2.2. Eligibility Criteria

Publications from 2010 to 2021 were searched to select appropriate studies using the following PCC framework: population (in vitro, animal, and human studies), concept (role of nHAp), and context (dentifrices/toothpaste containing nHAp as one of the ingredients). 

### 2.3. Search Strategy

An extensive search was conducted, on 31 November 2021, through the following literature databases: PubMed, Scopus, Web of Science Core Collection, and Cochrane Central Register of Controlled Trials. Keywords were included in the literature search. The search was conducted based on the two main concepts (nano-hydroxyapatite and dentifrices) of the research question. The literature database was searched using MeSH terms, keywords, and other free terms related to nano-hydroxyapatite and dentifrices. In addition, references of relevant studies and manual searching were also conducted for other potentially appropriate publications. Keyword searching of titles and abstracts was also performed with no barrier of date in the preliminary search. Table 1 shows the search strategy used in PubMed. The other databases and grey literature were also similarly searched.

Fifty-nine studies were found in the preliminary search, among which 11 studies were excluded during the title and abstract screening (Figure 1). Duplicate studies were excluded with the help of a citation/reference manager (name and version). The full texts of the remaining studies were examined by 2 reviewers, on the bases of the inclusion and exclusion criteria. A third reviewer was contacted in the case of a disagreement and resolved the differences through discussion, and a final consensus was reached to include 27 studies and exclude 18 studies. Table 2 shows the reasons for the excluded studies.

In this review, we included in vitro and in vivo studies, as well as human clinical trials, that used nHAp alone as one of the interventions. These studies were conducted on both permanent and primary teeth and on teeth with different stages and forms of caries and orthodontic bonding. However, case studies, case reports, reviews, editorials, or consensus papers, as well as studies with fewer than ten participants, were not included.

### 2.4. Data Charting and Items

Two reviewers selected the studies and charted the data and, if necessary, a third reviewer was consulted. For both in vitro and clinical investigations, data on the author’s name, the year of publication, the study’s objective, the study’s design, the samples assessed, the intervention/content of the dentifrice, the percentage of nHAp contained in the dentifrice, and the conclusions were charted. The duration of nHAp usage and its preventive use were also reported in clinical investigations.

### 2.5. Synthesis of Result

A qualitative synthesis of results was performed based on individual studies and is presented in the next section. 

## 3. Results

There were 48 studies identified using the first selection criteria, and when the exclusion and inclusion criteria were applied, these were further reduced to 28 studies (17 studies were in vitro investigations and 11 studies were clinical trials). There were no systematic reviews on the role of nHAp as a dentifrice. Only two reviews were found during our search regarding desensitizing toothpaste [22,23].

### 3.1. Characteristics of the Included In Vitro Studies 

Five in vitro studies employed bovine tooth samples, while the remaining 12 investigations used extracted human teeth (Table 3). Among them, two investigations [11,24] used human primary anterior teeth, while another employed human enamel slices from both primary and permanent teeth [25]. Human extracted premolars were the most frequently used permanent teeth in vitro, with only two studies utilizing human incisors [26,27]. While one in vitro study [5] assessed the effects of nHAp alone, the remaining studies compared the effects of nHAp alone to those of other components such as 20% zinc carbonate nHAp, 0.14 wt percent amine fluoride, 5% NovaMin (bioactive glass), and 8% Proargin. Various concentrations of nHAp were utilized in the in vitro investigations, ranging from 1% [28] to 20% [29,30]. The most often utilized concentration was 10% nHAp [5,11,13,14,24,25,29,31,32,33].

**Table 2 ijerph-19-05629-t002:** Excluded studies with reasons.

Author/Year	Type of Study	Conclusion	Reason for Exclusion
Sari et al. [34] 2021	In vitro	Remineralization of teeth and antibacterial/antibiofilm activity with nHAp and Curcuma aeruginosa toothpastes	nHAp was combined with Curcuma aeruginosa (C. aeruginosa)
Amaechi et al. [35] 2021	In situ	Remineralization and demineralization inhibition efficacies of nHAp dental lotion applied immediately after brushing teeth with nHAp toothpaste	5% nHAp dental lotion was used with 5% nHAp toothpaste
Ionescu et al. [36] 2020	In vitro	Decreased microbial colonization of RBC surfaces	Details of nHAp not clear
Kumar et al. [37] 2020	In vitro	Herbal dentifrice incorporated with nHAp had higher demineralizing potential as compared with a fluoride dentifrice	50% nHAp crystals were combined with herbal extract
Bologa et al. [38] 2020	In vitro	Dentinal tubules occluded and mineral deposition increased on the dentin surface with nHAp containing toothpastes	Details of nHAp not clear
Suryani et al. [39] 2020	Ex vivo	BAG and CCP-ACPF paste showed better remineralizing potential	Details of nHAp not clear
Wierichs et al. [40] 2020	In situ	Both fluoride-free dentifrices, one containing nHAp, did not hamper demineralization	Details of nHAp not clear
Alencar et al. [41] 2020	RCT	nHAP + PBM are effective in the control of dentin hypersensitivity	Details of nHAp not clear
Rajendran et al. [42] 2020	In vitro	Superior remineralization properties of Sr nHAp paste, found nontoxic	25 mol% Srn nHAp was used
Pei et al. [43] 2019	In vitro	nHAp-containing desensitizing toothpastes could occlude dentinal tubules. Application of nHAp desensitizers decreased bond strengths of the resin-dentin bonding.	Details of nHAp not clear
Alhamed et al. [44] 2019	Clinical study	nHAp was most effective in the treatment of initial carious lesion	Details of nHAp not clear
Reis et al. [45] 2018	In vitro	nHAp-containing dentifrice promoted less superficial roughness after 14 days	Details of nHAp not clear
Nozari et al. [46] 2017	In vitro	nHAp serum had remineralizing (initial caries) potential similar to NSF and NaF varnish	Details of nHAp not clear
Esteves-Oliveira et al. [47] 2017	In vitro	nHAp did not inhibit caries demineralization	20% Zinc-carbonate nHAp was used
Ebadifar et al. [48] 2017	In vitro	nHAp-containing toothpaste was more effective in remineralization	7% nHAp was combined with fluoride
Kamath et al. [49] 2017	In vitro	nHAp showed remineralization potential similar to others	Details of nHAp not clear
Ajami et al. [50] 2016	In vitro	Enamel surfaces and tooth color were not restored with nHAp serum	Details of nHAp not clear
Low et al. [51] 2015	Clinical study	Daily application of toothpaste containing potassium nitrate, sodium monoflurophosphate, and nHAp significantly reduced tooth pain due to dentin hypersensitivity	Details of nHAp not clear
Souza et al. [52] 2015	In situ	10% nHAp helps in remineralization	10% nHAp was combined with fluoride
Mielczarek and Michalik [53] 2014	In vitro	Reduction in surface roughness with nHAp but no significant improvement in SMH	1% nHAp was combined with 1450 ppm fluoride

SMH, surface microhardness; PBM, photobiomodulation; BAG, bioactive glass.

### 3.2. Characteristics of the Included Clinical Studies

The majority of clinical trials (nine) were RCTs, whereas two were in situ investigations (Table 3). The patient population ranged from 28 [57] to 105 [17,58] patients. These investigations were all conducted on adults aged 18 or older. The RCTs employed nHAp concentrations ranging from 1% to 25% [15,59]. The most often utilized concentration was 10% nHAp [16,17,57,60,61]. In the in situ investigations, the duration of nHAp use 1 min twice daily varied between 2 weeks and 28 days continuously [60,61]. In the RCT studies, the duration of 2 min twice daily applications ranged from 2 weeks [58] to 8 weeks [17]. Two trials showed daily use for 6 months [16,62]. Although it was widely used with brushing, nHAp was also applied using a cotton swab [57] and a custom-made tray [63].

### 3.3. Outcome of the Search Related to the Role of nHAp in Caries Preventive Applications

Nano-hydroxyapatite was reported to have prominent remineralization roles in many in vitro studies [11,13,14,25,26,27,28,33,56] and human clinical trials [62] (Table 4). It was also found that nHAp decreased caries susceptibility [32], enhanced enamel remineralization [32], inhibited caries [61], and reduced dentin demineralization [30]. Additionally, increased enamel remineralization was observed following orthodontic debonding with the application of 10% nHAp. [5]. However, few studies reported that nHAp reduced demineralization in the bovine enamel sample [29] or improved the tooth color of artificially created white spot lesions [31]. The dentinal desensitizing effect was seen in some in vitro studies [54,55], in situ studies [60], and the majority of the RCTs [15,17,58,59,60]. However, one study [63] reported an inferior result of 20% nHAp as compared with 20% pure silica. Wang et al. [57] concluded that nHAp lacked superior properties for reducing dentinal hypersensitivity. None of the RCTs evaluated the remineralization or desensitizing effects of nHAp in primary dentition. An in vitro study conducted in primary anterior teeth [11] reported better remineralizing effects with nHAp as compared with 1000 ppm fluoride, while another study [24] reported higher SMH values with NovaMin than with nHAp. In one study, caries such as lesions in primary teeth did not benefit from nHAp more than NovaMin [24]. Grewal et al. [25] demonstrated superior mineral gain and remineralization in enamel sections from primary and permanent teeth once nHAp had been used.

## 4. Discussion

The first nHAp dentifrice was marketed in Japan in 1993. Since then, several in vitro studies and clinical trials have been conducted using nHAp to evaluate its roles in caries prevention and remineralization. To the best of our knowledge, this is the first scoping review to explore and map the literature regarding the role of nHAp dentifrices. Our findings suggest that when nHAp is used in dentifrices, it plays a role in the remineralization of initial caries and the reduction in dentin demineralization, as well as in the reduction of white spot lesions and dentinal hypersensitivity.

### 4.1. Role of nHAp in Enamel Remineralization

Nano-hydroxyapatite is a biocompatible synthetic material similar to the hydroxyapatite crystals present in human teeth. It increases the degree of remineralization, especially in an acidic environment, by increasing the supply of calcium and phosphorus ions to the demineralized zone [13]. Its therapeutic role in tooth remineralization has been widely studied. We identified studies reporting that nHAp played a significant role in remineralization in vitro [11,13,14,25,26,27,28,33,56] and in human clinical trials [62]. The basis of application of nHAp is that the balance between demineralization and remineralization is controlled by the salivary saturation of apatite minerals [64], and enhancing the salivary levels of calcium and phosphate concentrations appears to be a potential method to increase remineralization in teeth and to inhibit demineralization [65]. Nano-hydroxyapatite has also been shown to increase elevated surface energy, to increase atomicity, and to exhibit strong bonding to enamel surfaces [56]. Increased enamel remineralization after orthodontic debonding has also been found [5]. Since nHAp is most effective when used 2–3 min daily for at least 10 days/month, a bi-weekly, twice a day regimen following orthodontic treatment is recommended to reduce post-debonding roughness and sensitivity [5]. Nano-hydroxyapatite has been found to increase tooth brightness [32] by remineralization of the enamel surface and the closure of gaps in the enamel surface that inhibits bacterial growth. Thus, nHAp is a remineralizing agent as well as a healthier whitening option. Further studies are warranted to confirm the durability of this result. However, a few studies have reported that nHAp did not improve the tooth color of artificially created white spot lesions [31] and did not reduce demineralization in the bovine enamel sample [29]. This could be because the highly remineralized outer enamel surface blocked the diffusion of mineral ions into deeper areas of enamel, thus restricting enamel recrystallization in deeper areas [53]. Different modes of nHAp application such as constant use of nHAp for 28 days [61] and daily use for 6 months [62] could also cause variations in the final results.

### 4.2. Role of nHAp in Enamel Demineralization

Nano-hydroxyapatite can provide a source of calcium for the oral cavity; increased calcium levels can help to limit the acid challenge, reducing enamel demineralization while promoting enamel remineralization. This calcium phosphate reservoir may contribute to an enamel mineral oversaturation state, hence decreasing demineralization and enhancing remineralization [13,66]. It has also been found that nHAp decreased caries susceptibility [32], enhanced enamel remineralization [32], inhibited caries [61], and reduced dentin demineralization [30]. Possibly, this could be due to the deposition of a new homogenous apatite surface layer on the demineralized surface [31]. This mechanism protects the underlying diseased surface from further demineralization and promotes remineralization [13]. Nano-hydroxyapatite also promotes more minerals to be accumulated in the outer layer of carious lesions, thus resulting in a highly mineralized external layer and inhibiting mineral ions from entering deeper regions of the demineralized lesion [13].

### 4.3. Role of nHAp in Reducing Dentinal Sensitivity

A dentinal desensitizing effect was another significant action of nHAp dentifrices, which was seen in some in vitro studies [54,55], in situ studies [60], and the majority of the RCTs [15,17,58,59,60]. The most common therapeutic method to treat dentinal hypersensitivity and related pain is by occlusion of the exposed dentin tubules to decrease dentin permeability and prevent rapid movement of fluid. Nano-hydroxyapatite forms a protective layer on the external surface of dentin in human root specimens, resulting in the occlusion of dentin tubules [30] by mineral hydroxyapatite, thus reducing dentinal permeability, and preventing fluid disturbance within the tubules and decreasing dentinal hypersensitivity [60]. Furthermore, nHAp-containing toothpaste has been found to increase salivary calcium concentrations [67] and obliterate micropores on tooth surfaces [32,68]. Hence, nHAp particles attract an enormous amount of calcium and phosphate ions from the surrounding solutions (saliva, dentifrices, and mouth rinses) to the tooth tissue, thus promoting crystal integrity and growth [68]. However, contradictory results [63] have shown that 20% pure silica, when used as a positive control, was more effective than 20% nHAp in reducing dental pain scores. This could be because none of the clinical studies that reported higher effectiveness of nHAp included a positive control group. Additionally, the four-point dental pain scale’s restrictive nature limited participants’ abilities to categorize their discomfort. However, the percentage change from baseline was lower at each time point when they were describing having pain [63]. None of the studies reported a standardized duration for the application of nHAp. While a short duration (1 min twice daily for 2 weeks) was reported in an in situ study [60], the RCT studies reported applications of nHAp for 2 min a day twice daily from 2 weeks [58] to 8 weeks [17]. Although most studies applied nHAp with a toothbrush, a cotton swab [57] and custom-made tray [63] were also used.

### 4.4. The Optimal Concentration of nHAp

The most frequently utilized concentration in the RCTs was 10% nHAp [16,17,57,60,61]. The same concentration was used in the in vitro studies [5,11,13,14,24,25,29,31,32,33]. The rate and amount of nHAp precipitation increased with higher concentrations, as there was an increase in the deposition of calcium and phosphate ions [69], as well as the surface hardness of the demineralized enamel [70]. Although 15% nHAp demonstrated efficient remineralization, this concentration was too high for practical usage in mouthwash or toothpaste, as concentrations in this range would inevitably generate some level of aggregation [69]. Moreover, 10% nHAp showed similar results as compared with 15% nHAp; therefore, a 10% nHAp formulation appears to be an optimal concentration for remineralization of early enamel caries. It is also noteworthy that the remineralization rate was the fastest during the first 6 days of pH cycling, and then it slowed and stabilized beyond this point [71]. With concentrations below 10%, nHAp exerted an increased remineralization effect, and there was a sharp change in this trend when the concentration was between 5% and 10% [69].

### 4.5. Limitations

This is the first review study to consolidate the existing literature on the effects of adding nHAp to dentifrice. The review analysis omitted studies that did not clearly characterize nHAp formulation. In addition, the source of samples, the duration of the study, the mode and duration of nHAp administration, the maturity of the tooth structure, and the brittleness of the tooth sample all have the potential to affect the outcomes. In vitro environments do not always accurately replicate the natural oral environment, and the results cannot always be immediately applicable to clinical case scenarios. Similarly, the pH cycling used in in vitro experiments to mimic de-/remineralization processes does not accurately reflect the oral environment. The outcome of this scoping review stresses the importance of more clinical research to obtain more reliable information.

### 4.6. Future Direction

There is a need to standardize the procedure and to provide additional information about the use of nHAp dentifrices. Recent research has reported conflicting findings when nHAp was combined with strontium (Rajendran et al., 2020), *Salvadora persica* [37], ozone therapy [16], photobiomodulation [41], and dental lotions [35]. Further research is needed to determine the effects of nHAp on periodontal tissues and fibroblastic growth. Additionally, the long-term effects of various laser wavelengths and power settings on dentin hypersensitivity must be investigated. Another area of inquiry appears to be the antibacterial activity of herbal extracts added into nHAp dentifrices. Adjunctive periodontal therapy, including ozone therapy and photobiomodulation, should be explored in conjunction with nHAp dentifrices for remineralization, desensitization, tooth whitening, and preventative benefits. Larger multi-arm studies that evaluate the clinical and patient-centered outcomes of nHAp may be undertaken in the future. Additionally, research examining the cost-effectiveness and patient-centered outcomes of nHAp should be conducted in a broader segment of the population, including uncooperative children and/or patients.

## 5. Conclusions

According to this scoping review, nHAp has a variety of beneficial effects when used in dentifrices, including increased remineralization in initial enamel lesions, caries inhibition, decreased demineralization, increased tooth brightness, decreased dentinal hypersensitivity and associated pain, decreased surface roughness, and remineralization of enamel following orthodontic debonding. The optimal concentration of nHAp in dentifrices is 10%. Currently, the evidence supporting the efficacy of nHAp dentifrices in primary teeth is limited. Additional long-term studies employing standardized protocols are necessary to draw definitive findings about the effect of nHAp dentifrices in primary and permanent dentitions.

## Figures and Tables

**Figure 1 ijerph-19-05629-f001:**
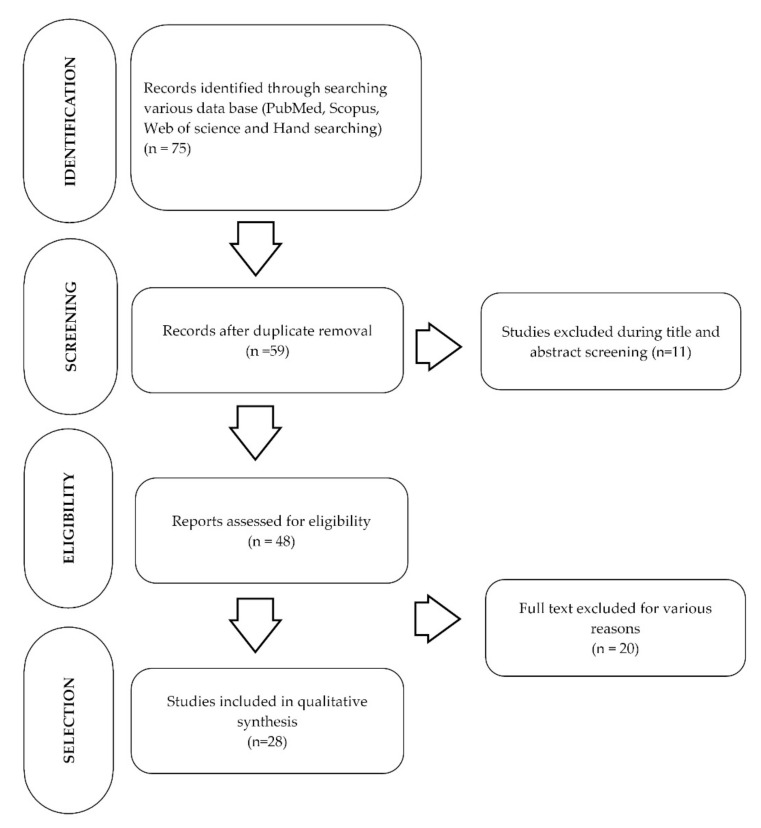
PRISMA flow diagram showing the selection of articles included in the review.

**Table 1 ijerph-19-05629-t001:** Search Strategy for PubMed.

Search Strategy	MeSH/Keywords	Result
#1	((“nano” [Journal] OR “nano” [All Fields]) AND (“durapatite” [MeSH Terms] OR “durapatite” [All Fields] OR “hydroxyapatite” [All Fields] OR “hydroxyapatites” [MeSH Terms] OR “hydroxyapatites” [All Fields])) AND (2000:2022 [pdat])	2423
#2	(“dentifrices” [Pharmacological Action] OR “dentifrices” [MeSH Terms] OR “dentifrices” [All Fields] OR “dentifrice” [All Fields]) AND (2000:2021 [pdat])	7839
#3	“toothpastes” [All Fields] OR “toothpastes” [MeSH Terms] OR “toothpastes” [All Fields] OR “toothpaste” [All Fields]	6022
#4	#2 OR #3	9751
#5	#1 AND #4	59
	Title and abstract screening	48
	Full text with inclusion and exclusion criteria	28
	Excluded studies	20

**Table 3 ijerph-19-05629-t003:** Summary of publications that reported the effect of nano-hydroxyapatite dentifrice in in vitro studies.

Author/ Year	Aim	Samples Assessed	Percentage of nHAp	Conclusion
Verma and Pandian [5] 2021	Effects of nHAp dentifrice on demineralized surface of enamel post orthodontic debonding	Maxillary premolars, therapeutic extraction	10% nHAp	Superior remineralizing effect of nHAp dentifrice on enamel after post- orthodontic debonding
Juntavee et al. [14] 2021	Remineralization effect of various non-fluoridated and fluoridated toothpaste	Extracted human premolars	10% nHAp	nHAP improved remineralization for treating initial carious lesions
Kasemkhun and Rirattanapong, [11] 2021	Remineralizing effect of various non-fluoridated toothpastes on artificial caries in primary teeth	Intact primary incisor teeth	10% nHAp	Better remineralizing of primary teeth with nHAp than 1000 ppm fluoridated toothpaste
Geeta et al. [26] 2020	Remineralizing effect of four agents on initial enamel lesion	Human maxillary central incisors	1% nHAp	nHAp-containing dentifrice has highest remineralizing potential
Hammad et al. [31] 2020	Color changes and stability of the resin infiltrant on WSLs with nHAp	Enamel surfaces with artificially created WSLs	10% nHAp	Better color change of WSLs with resin infiltrant than nHAp toothpaste
Leal et al. [30] 2020	Effectiveness of high fluoride and nHAp-containing dentifrice on root dentin demineralization.	Dentin specimens were obtained from bovine incisors	20% nHAp	nHAp reduced dentin demineralization
Manchery et al. [33] 2019	Remineralization ability of nHAp, NovaMin, and amine fluoride dentifrice on artificial enamel caries	Extracted sound premolars	10% nHAp	nHAp can remineralize artificial carious lesions
Joshi et al. [28] 2019	Estimate initial stage of demineralization through remineralization potential of four commercially available agents	Permanent intact premolar	1% nHAp	nHAp improved remineralization and SMH
Grewal et al. [25] 2018	Remineralizing efficacy of the three dentifrices on both primary and permanent enamel surfaces	Enamel sections from primary and permanent molars	10% nHAp	nHAp exhibited highest remineralization (mineral gain)
Jena et al. [54] 2017	SEM study of dentinal tubule occlusion using four different desensitizing dentifrices	Dentin blocks from human molars	15% nHAp	Reduction in dentinal hypersensitivity with 15% nHAp
Haghgoo et al. [24] 2016	Remineralizing of primary tooth caries-like lesions with NovaMin and nHAp	Sound human primary anterior teeth	10% nHAp	Equal effect of nHAp and NovaMin in caries like lesions of primary teeth remineralization
Kulal et al. [55] 2016	Dentinal permeability and tubule occlusion with 15% nHAp	Dentin specimen premolars	15% nHAp	nHAp exerts desensitizing effect
Vyavhare et al. [27] 2015	Effect of nHAp on remineralization of early carious lesions	Initial artificial caries in maxillary incisors	10% nHAp	nHAp can remineralize initial enamel lesions
Comar et al. [29] 2013	Preventive potential of experimental nHAp pastes with or without fluoride	Bovine enamel	10% and 20% nHAp	No effect of nHAp in reducing dental demineralization in vitro.
Tschoppe et al. [56] 2011	Effects of nHAp toothpastes on remineralization	Bovine enamel and dentin subsurface lesions	7 wt.% pure nHAp	Toothpastes containing nHAp showed better remineralization than amine fluoride toothpastes
Huang et al. [13] 2011	Artificial enamel caries remineralization effect of nHAp	Demineralized bovine enamel	10% nHAp	Good remineralizing potential of nHAp in initial enamel caries
Hwang et al. [32] 2010	Effect of nHAp on remineralization	Bovine tooth	15% nHAp	Dentifrices with nHAp increased in brightness, enamel remineralization, decrease in caries susceptibility

CPP-ACP, casein phosphopeptide-amorphous calcium phosphate; DH, dentinal hypersensitivity; SEM, scanning electron microscope; WSLs, white spot lesions.

**Table 4 ijerph-19-05629-t004:** Summary of publications that reported effects of nano-hydroxyapatite dentifrices in human clinical studies.

Author/Year	Aim	Study Design/Population	Duration of Application	Percentage of nHAp	Conclusion
Amaechi et al. [17] 2021	DH reduction using nHAp and CSPS	RCT/18–80-year-old subjects with DH, permanent teeth	2 min twice a day for 8 weeks	10% and 15% nHAp	Toothpaste containing nHAp (10 or 15%) DH symptoms
Grocholewicz et al. [16] 2020	Remineralization of initial approximal caries using three methods	RCT/92 patients between 20–30 years of age	6 Months daily use	10% of nHAp	Improved remineralization when nHAp gel and ozone therapy were combined
Badiee et al. [62] 2019	Remineralization of early enamel lesions in nHAp dentifrice users	RCT/50 patients on fixed orthodontic treatment	Twice daily for 6 months	6.7% nHAp	Better remineralization and reduction in extent of lesion with nHAp toothpaste
Vano et al. [15] 2018	Efficacy of nHAp toothpaste compared to fluoride in reducing DH	RCT/105 subjects	Twice daily for 4 weeks	25% nHAp	nHAp fluoride free toothpaste is effective, reduces DH
Amaechi et al. [63] 2018	Reduction in DH with nHAP dental cream and pure silica	RCT/51 subjects aged 18 to 80 years	5-min application once daily	20% nHAp	Both showed similar relief for DH, but silica reduced dental pain score better than nHAp
Anand et al. [2] 2017	nHAp toothpaste in the management of DH	RCT/30 patients in each group (2 groups)	1-min application/brushing 2 min twice	1% nHAp	DH decreased with nHAp
Wang et al. [57] 2016	Desensitizing effect of nHAp	RCT/28 subjects with 137 teeth	Twice a day 4 min with cotton swab	10% and 20% nHAp	nHAp was effective in reducing dentin hypersensitivity
Amaechi et al. [60] 2015	Comparison of dentin tubule occlusion by different toothpaste	In situ/80 participants	1 min twice a day for 2 weeks	10% and 15% nHAp	nHAP more effectiveness in occluding dentin tubules
Gopinath et al. [59] 2015	Effectiveness of nHAp in reducing DH	RCT/36 patients	2 min twice a day for 4 weeks	1% nHAp	nHAp reduced DH
Vano et al. [58] 2014	Efficacy in reducing DH with nHAp	RCT/105 subjects	2 min twice a day for 2 weeks/4 weeks	15% nHAp	nHAp toothpaste effective as a desensitizing agent
Najibfard et al. [61] 2011	Efficacy of nHAp dentifrices on caries remineralization and demineralization inhibition	In situ/30 adults in four-phase study lasting 28 days	28 Days constantly	5% and 10% nHAp 10% nHAp	Remineralization and inhibition of caries occurred with nHAp dentifrice

CSPS, calcium sodium phosphosilicate; DH, dentinal hypersensitivity.

## Data Availability

The datasets used and/or analyzed during the current study are available from the corresponding author on reasonable request.

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
