# Peer review of "Nano-Hydroxyapatite (nHAp) in the Remineralization of Early Dental Caries: A Scoping Review"

_ijerph, 2022, doi:10.3390/ijerph19095629_

Round 1
Reviewer 1 Report
See attached pdf document (Reviewer report).

Author Response
Reviewer 1
Comments and Suggestions for Authors
See attached pdf document (Reviewer report).
Submission Date
Reviewer Report
Manuscript ID: ijerph-1677683
MS Title: Nano-hydroxyapatite (nHAp) in the remineralization of early dental caries: a scoping review
General comment.
This manuscript reported a scoping systematic review of literature on the various effects of nanoHydroxyapatite (nHAP) when it is the sole active ingredient in toothpaste and used alone. This review is timely and of high clinical importance, considering the growing interest in nHAP-containing oral care products. Although the review was well conducted, the report was poorly organized and a bit difficult to read due to poor grammar constructions.
The manuscript has been substantially revised and the language has been edited by the mdpi editing service. Attached is a certificate of editing.
Specific comments
- The objective of the review in the abstract is different from that in the main text. In abstract, the authors indicated that the objective was specifically to analyze the remineralization potential of nHAp) containing dentifrices, but in the main text they went more general that the objective is analyze the role of nHAp in dentifrices. Reading the manuscript, it was obvious that the analysis was not only on remineralization, I advise that the objective in the abstract be amended to be similar that in text.
The manuscript is edited to emphasize the study's primary objective.
- Actually, the two objectives in the text is conflicting. I would suggest you make the objective more concise and clear with one statement. For example, “ to systematically examine and map the existing research on the roles of nHAp in dentifrices when used alone”. I believed that this one statement covers the therapeutic, preventive, and esthetic roles of nHAP in dentifrice, which are the review was reporting.
The objective and research question have been rewritten to concentrate exclusively on the effect of nHAp on enamel remineralization.
- Some statements in ‘Introduction’ need references. For example:
- Page 1, line 35; Dental caries is the most common oral illness, affecting people of all ages.
Kassebaum, N.J.; Bernabé, E.; Dahiya, M.; Bhandari, B.; Murray, C.J.; Marcenes, W. Global burden of untreated caries: A systematic review and metaregression. J Dent Res 2015, 94, 650-658.
- Page 1, line 36,; It raises the expense of therapy and economic burden on both individuals and healthcare systems.
Kastenbom, L.; Falsen, A.; Larsson, P.; Sunnegårdh-Grönberg, K.; Davidson, T. Costs and health-related quality of life in relation to caries. BMC Oral Health 2019, 19, 187.
- Page 2, line 50; “ , as a result of continuous use of fluoride concentrations”. I don’t understand what the authors mean by this segment of this statement. Do you mean ‘as a result of continuous use of high fluoride concentrations? Please clarify.
There is a typographical error. Now modified
as a result of continuous use of high fluoride concentrations
- Page 7, line 156; “3.2. Characteristics of the included clinical studies (Table 3)”. This should be Table 4, not Table 3.
Corrected
- Page 7, lines 166 and 178. I think splitting the objective of the review into two conflicting aims is confusing and led to repetitions in sections 3.3 and 3.4. I suggest to merger section 3.3 amd 3.4.
The section removed and merged.
- In ‘Discussion’ section, the role of nHAp in enamel demineralization should be separated from section 4.1 (Role of nHAp in enamel remineralization), and discussed as a separate role.
An additional section on enamel demineralization added.
- Page 9, line 209. “It was also found that nHAp decreased caries susceptibility [30], enamel remineralization [30], inhibited caries [36], and reduced dentin demineralization [28]” Does this statement meant that nHAP decreased enamel remineralization?
The sentence corrected as follows.
It was also found that nHAp decreased caries susceptibility [32], enhanced enamel remineralization [32], inhibited caries [61], and reduced dentin demineralization [30].
- Page 9, lines 213-216. “nHAp also promotes more minerals to be accumulated in the outer layer but inhibits mineral ions from entering deeper regions of the demineralized lesion [11], thus resulting in a highly mineralized external layer.” Please rephrase this statement to “nHAp also promotes more minerals to be accumulated in the outer layer of carious lesion, thus resulting in a highly mineralized external layer, inhibits mineral ions from entering deeper regions of the demineralized lesion.”
Section 4.4 is an exact repetition of section 3.4. ‘Discussion’ should not be a repetition of ‘Result’.
The section removed as recommended.

Reviewer 2 Report
I read the article in depth. Looks interesting and well written.
- Please add PRISMA details
- Please add any flow diagram/illustration
- Please go for SR and meta-analysis.
- Please add forest plot analysis.
Author Response
Comments and Suggestions for Authors
I read the article in depth. Looks interesting and well written.
- Please add PRISMA details
PRISMA details added as Appendix 1.
- Please add any flow diagram/illustration
Flow Diagram added as Figure 1
- Please go for SR and meta-analysis. Please add forest plot analysis.
Since the results / Observations of the various studies reviewed were inconsistent it is beyond the scope of the current review.

Reviewer 3 Report
This study is a scoping review study. The authors define precisely the research question. The design of the study was well-conducted. However, a native English speaker needs to revise.
Author Response
Comments and Suggestions for Authors
This study is a scoping review study. The authors define precisely the research question. The design of the study was well-conducted. However, a native English speaker needs to revise.
The manuscript has been substantially revised and the language has been edited by the mdpi editing service. Attached is a certificate of editing.
